# Induced Pluripotent Stem Cells in Cardiomyopathy: Advancing Disease Modeling, Therapeutic Development, and Regenerative Therapy

**DOI:** 10.3390/ijms26114984

**Published:** 2025-05-22

**Authors:** Quan Duy Vo, Kazufumi Nakamura, Yukihiro Saito, Satoshi Akagi, Toru Miyoshi, Shinsuke Yuasa

**Affiliations:** 1Department of Cardiovascular Medicine, Okayama University Graduate School of Medicine, Dentistry and Pharmaceutical Sciences, Okayama 700-8558, Japan; dr.duyquan@gmail.com (Q.D.V.); akagi-s@cc.okayama-u.ac.jp (S.A.); miyoshit@cc.okayama-u.ac.jp (T.M.); yuasa@okayama-u.ac.jp (S.Y.); 2Center for Advanced Heart Failure, Okayama University Hospital, Okayama 700-8558, Japan; 3Department of Cardiovascular Medicine, Okayama University Hospital, Okayama 700-8558, Japan; p5438a3l@s.okayama-u.ac.jp

**Keywords:** induced pluripotent stem cells, cardiomyopathy, disease modeling, drug screening, regenerative therapy

## Abstract

Cardiomyopathies are a heterogeneous group of heart muscle diseases that can lead to heart failure, arrhythmias, and sudden cardiac death. Traditional animal models and in vitro systems have limitations in replicating the complex pathology of human cardiomyopathies. Induced pluripotent stem cells (iPSCs) offer a transformative platform by enabling the generation of patient-specific cardiomyocytes, thus opening new avenues for disease modeling, drug discovery, and regenerative therapy. This process involves reprogramming somatic cells into iPSCs and subsequently differentiating them into functional cardiomyocytes, which can be characterized using techniques such as electrophysiology, contractility assays, and gene expression profiling. iPSC-derived cardiomyocyte (iPSC-CM) platforms are also being explored for drug screening and personalized medicine, including high-throughput testing for cardiotoxicity and the identification of patient-tailored therapies. While iPSC-CMs already serve as valuable models for understanding disease mechanisms and screening drugs, ongoing advances in maturation and bioengineering are bringing iPSC-based therapies closer to clinical application. Furthermore, the integration of multi-omics approaches and artificial intelligence (AI) is enhancing the predictive power of iPSC models. iPSC-based technologies are paving the way for a new era of personalized cardiology, with the potential to revolutionize the management of cardiomyopathies through patient-specific insights and regenerative strategies.

## 1. Introduction

Cardiomyopathies represent a heterogeneous group of myocardial disorders characterized by structural and functional abnormalities of the heart muscle, often resulting in heart failure, arrhythmias, thromboembolic events, and an increased risk of sudden cardiac death [1,2]. The classification of cardiomyopathies has undergone significant transformation over the decades, reflecting the growing understanding of underlying pathophysiology. In the 1970s, Goodwin and Oakley provided one of the earliest formal definitions, describing cardiomyopathies as “diseases of the heart muscle of unknown origin”, thereby distinguishing them from ischemic, hypertensive, and valvular heart diseases. They introduced a functional classification that grouped cardiomyopathies into three primary forms: (i) congestive cardiomyopathy, characterized by impaired systolic function; (ii) hypertrophic cardiomyopathy, with or without left ventricular outflow tract obstruction, associated with abnormal diastolic filling; and (iii) obliterative or constrictive cardiomyopathy, defined by restrictive ventricular physiology [3].

In recent years, the integration of molecular biology, genetic testing, and high-resolution cardiac imaging has significantly advanced our understanding of the genetic architecture, pathophysiological mechanisms, and clinical variability of cardiomyopathies. In response to these advancements, the Padua classification has been proposed as a contemporary and comprehensive framework. Unlike earlier systems that focused solely on morphology or function, the Padua classification incorporates etiological insights—such as genetic mutations and molecular pathology—alongside phenotypic expression, including morpho-functional and structural remodeling of the ventricles. This integrative approach categorizes cardiomyopathies into three principal groups: hypertrophic/restrictive cardiomyopathy (H/RC), dilated/hypokinetic cardiomyopathy (D/HC), and scarring/arrhythmogenic cardiomyopathy (S/AC) [4].

Within this classification, several clinically important cardiomyopathy subtypes are recognized, each with distinct genetic and phenotypic characteristics. Hypertrophic cardiomyopathy (HCM) is the most common inherited form, affecting approximately 1 in 500 individuals, and is primarily caused by mutations in sarcomeric genes such as *MYH6* (OMIM: 160710), *MYH7* (OMIM: 160760), and *MYBPC3* (OMIM: 600958) [5,6,7,8]. Dilated cardiomyopathy (DCM), with an estimated prevalence of 1 in 2500, may arise from genetic mutations—including *TTN* (OMIM: 188840) and *LMNA* (OMIM: 150330)—as well as acquired causes such as viral myocarditis, toxic exposures, or autoimmune processes. DCM remains a leading indication for cardiac transplantation [9,10,11].

Restrictive cardiomyopathy (RCM), although relatively rare—accounting for only 2–5% of all cardiomyopathies—is characterized by severe diastolic dysfunction and often indicates a poor prognosis [12]. The genetic basis of RCM includes pathogenic variants in sarcomeric genes such as *TNNI3* (OMIM: 191044), *TNNT2* (OMIM: 191045), *TNNC1* (OMIM: 191040), *TPM1* (OMIM: 191010), *ACTC1* (OMIM: 102540), *MYL2* (OMIM: 160781), *MYL3* (OMIM: 160790), *MYH7*, *MYBPC3*, and *TTN* [13,14,15,16,17,18]. In addition, non-sarcomeric genes implicated in RCM pathogenesis include *LMNA*, *DES* (OMIM: 125660), *FLNC* (OMIM: 102565), *ACTN2* (OMIM: 102573), *MYPN* (OMIM: 608517), *CRYAB* (OMIM: 123590), *BAG3* (OMIM: 603883), *TMEM87B* (OMIM: 617919), and *DCBLD2* (OMIM: 618404) [19,20,21,22,23,24]. Mutations in these genes disrupt myocardial architecture and cellular integrity, culminating in the diastolic dysfunction characteristic of RCM.

Arrhythmogenic cardiomyopathy (ACM) is a primary myocardial disorder that occurs independently of ischemic, hypertensive, or valvular heart diseases [25]. Although this phenotype was initially believed to predominantly affect the right ventricle, accumulating evidence has shown that the left ventricle may be equally involved, or in some cases, may even represent the predominant site of disease [26,27]. The genetic landscape of ACM is diverse, with approximately half of cases linked to mutations in desmosomal genes such as *PKP2* (OMIM: 602861), *DSP* (OMIM: 125647), *DSG2* (OMIM: 125671), *DSC2* (OMIM: 125645), and *JUP* (OMIM: 173325), which encode components of the cardiac intercalated disc [28,29,30,31,32]. Additionally, mutations in non-desmosomal genes—*FLNC, CTNNA3* (OMIM: 607667), and *TMEM43* (OMIM: 612048)—have expanded the spectrum of disease-related pathways, including those involved in cytoskeletal integrity, nuclear envelope structure, and cell–cell adhesion [33,34,35]. ACM affects an estimated 1 in 1000 to 1 in 5000 individuals, and sudden cardiac death (SCD) is often the first clinical manifestation, with post-mortem studies attributing 20–31% of SCD cases to ACM [36,37].

Despite significant advances in our understanding of these cardiomyopathy subtypes, traditional research models present substantial limitations in replicating the complex pathophysiology observed in humans. Animal models and heterologous cell systems often fail to replicate human cardiac physiology and disease manifestations. For example, mice have fundamental differences in cardiac electrophysiology and may not exhibit the same phenotypes seen in human mutations [38]. Moreover, patients with identical genetic mutations can show variable disease severity due to genetic background and environmental modifiers—phenomena difficult to capture in non-human models [39,40]. Primary human cardiomyocytes from patient biopsies are scarce and cannot be expanded in culture. These challenges have created a need for patient-specific disease models that accurately reflect human cardiomyopathy. In this context, induced pluripotent stem cells (iPSCs) have emerged as a game changer in cardiac research. First pioneered by Yamanaka et al. in 2006, iPSC technology involves reprogramming adult somatic cells into pluripotent stem cells by introducing a set of transcription factors (OCT4, SOX2, KLF4, c-MYC) [41]. This breakthrough allows researchers to create pluripotent cell lines from any individual, which can then be differentiated into virtually any cell type, including cardiomyocytes.

iPSCs provide an unlimited source of patient-derived cardiomyocytes in vitro, capturing the patient’s unique genetic makeup. iPSC-derived cardiomyocytes (iPSC-CMs) reflect many aspects of the donor’s cardiac cell biology, offering a “disease in a dish” model for cardiomyopathies. Unlike immortalized cell lines, iPSC-CMs express the full complement of cardiac ion channels, receptors, and structural proteins, making them more physiologically relevant for studying drug responses and disease mechanisms [42,43]. Patient-specific iPSC-CMs have been shown to recapitulate key phenotypes of inherited cardiomyopathies and channelopathies, helping to unravel how specific mutations lead to cellular dysfunction [44,45]. By inheriting the patient’s exact genotype, iPSC-CMs can also mirror idiosyncratic features of the disease, addressing issues of incomplete penetrance and variable expressivity seen in patients. These properties make iPSC-CMs a powerful platform for both mechanistic studies and translational applications such as testing personalized therapies.

In this article, we review the state of the art in using iPSCs for cardiomyopathy research and therapy development. Through this comprehensive overview, we aim to illustrate how iPSC technology is driving a paradigm shift in cardiomyopathy research, paving the way toward personalized and regenerative cardiology.

## 2. Generation of iPSCs for Cardiomyopathy Models

### 2.1. Reprogramming Somatic Cells into iPSCs

The creation of iPSCs from adult somatic cells is the foundational step for generating patient-specific cardiomyocytes. Takahashi and Yamanaka’s landmark work demonstrated that introducing a defined set of embryonic transcription factors (the Yamanaka factors: OCT3/4, SOX2, KLF4, and c-MYC) into differentiated cells can induce pluripotency. In the original 2006 experiments, mouse fibroblasts were successfully reprogrammed into iPSCs [41], and this was soon replicated with human adult fibroblasts using the same four-factor cocktail [46]. These iPSCs exhibit the key properties of pluripotent stem cells: they can self-renew indefinitely and differentiate into cell types of all three germ layers [41]. Practically, reprogramming can be achieved from various accessible tissues such as skin biopsies (fibroblasts) [46,47], blood collection (peripheral blood mononuclear cells) [48,49,50], or even urine samples [51,52], making the process clinically feasible.

Several methods are available for delivering reprogramming factors, each with advantages for potential therapeutic use. Traditional protocols used integrating viral vectors (e.g., retroviruses or lentiviruses) to deliver the Yamanaka genes, which is efficient but leaves behind proviral integrations in the genome [53]. To avoid insertional mutagenesis and residual transgene expression, integration-free methods have been developed. These include non-integrating Sendai viruses (which are RNA viruses that do not enter the host genome), episomal plasmid vectors, synthetic mRNAs, or small molecules to activate endogenous pluripotency networks [54,55]. Quality control is crucial once iPSC lines are established. Clonal iPSC lines are expanded and screened for hallmarks of pluripotency: expression of pluripotency markers (such as OCT4, NANOG, SSEA-4, and TRA-1-60), the ability to form teratomas containing tissue derivatives of all germ layers, and a normal karyotype. Only well-characterized iPSC lines are advanced for differentiation to ensure the reliability of cardiomyopathy models [56,57].

### 2.2. Differentiation into Cardiomyocytes

To study cardiomyopathies, iPSCs must be efficiently differentiated into functional cardiomyocytes. Early differentiation protocols for human iPSCs were adapted from embryonic stem cell methods, often relying on embryoid body formation and the addition of growth factors (activin A, BMP4) to induce cardiac mesoderm and cardiogenesis [58,59]. However, these approaches yielded variable and often low cardiomyocyte purity. Over the past decade, more defined and robust protocols have been developed by manipulating developmental signaling pathways, particularly the Wnt/β-catenin pathway [60]. In brief, a biphasic Wnt modulation strategy is now commonly used: iPSC cultures are first treated with a Wnt signaling activator (such as GSK3β inhibitor) for a short period to induce mesodermal differentiation, and subsequently treated with a Wnt signaling inhibitor to drive specification into cardiac progenitors and cardiomyocytes [61,62]. Differentiation is typically confirmed by the appearance of spontaneously beating cell clusters around 7–14 days of induction, reflecting the development of functional syncytial cardiomyocytes. Modern protocols are serum-free and chemically defined, improving reproducibility and compatibility with clinical-grade production [63,64].

Recent insights have highlighted a novel role of progesterone and its receptor (PR) during the reprogramming and early differentiation of iPSCs. While progesterone has long been recognized for its regulatory role in cardiac development and metabolic programming, Manganelli et al. (2024) demonstrated for the first time that PR is constitutively expressed in human iPSC lines derived from both CD34^+^ hematopoietic progenitors and dermal fibroblasts, regardless of the reprogramming strategy employed [65]. The presence of PR in the nuclear compartment of iPSCs, confirmed by both immunofluorescence and flow cytometry, suggests that progesterone signaling may influence transcriptional programs even before lineage-specific differentiation begins. These findings open new perspectives for understanding the impact of steroid hormone signaling in early cardiac differentiation and may provide a foundation for enhancing maturation strategies of iPSC-derived cardiomyocytes by targeting progesterone pathways [66].

In two-dimensional monolayer cultures, iPSC-derived cardiomyocytes usually begin beating by 1–2 weeks and can be maintained for months. Alternative methods, such as three-dimensional cardiac organoids or engineered heart tissue, can also be used to generate cardiomyocytes in a more native-like microenvironment [67,68]. After differentiation, various enrichment techniques can further improve cardiomyocyte purity. One common approach is metabolic selection: exploiting the unique ability of cardiomyocytes to survive in lactate-containing, glucose-depleted media [69,70]. Flow cytometry or magnetic sorting for cardiomyocyte-specific surface markers is another strategy to isolate iPSC-CMs to high purity for characterization or therapy [71,72].

### 2.3. Characterization of iPSC-Derived Cardiomyocytes

Thorough characterization of iPSC-derived cardiomyocytes is essential to validate them as a model for true cardiac muscle cells, especially given their relatively immature state.

Molecular and structural markers are assessed to confirm cardiac lineage. Immunostaining and gene expression analyses should demonstrate robust expression of cardiomyocyte-specific proteins such as cardiac troponin T (*TNNT2*), sarcomeric α-actinin (*ACTN2*), myosin heavy chains (*MYH6* and *MYH7*), and transcription factors like NKX2-5 [73]. By immunofluorescence, iPSC-CMs typically exhibit organized sarcomeric striations (though less organized than adult myocytes), indicating assembly of the contractile apparatus [74]. Ultrastructural examination (electron microscopy) can reveal developing myofibrils and junctions, although features like mature intercalated discs and transverse tubules are largely absent in early-stage iPSC-CMs [75,76].

Functional characterization is particularly important to establish that iPSC-CMs behave like cardiomyocytes. Electrophysiological profiling can be done via patch-clamp recordings or multi-electrode array (MEA) systems [77]. iPSC-CMs exhibit action potentials that can be classified as nodal-like, atrial-like, or ventricular-like based on their shape and duration. Calcium handling can be assessed with fluorescent calcium indicators, demonstrating Ca2+ transients that coincide with contractions [78]. While iPSC-CMs do contract spontaneously, their contractile force and kinetics can be quantified using methods like traction force microscopy, micropost arrays, or by forming engineered microtissues to measure tension. Typically, their contraction amplitudes and force output are much smaller than those of adult cardiomyocytes, reflecting an immature phenotype [79,80] (Figure 1).

## 3. Applications of iPSCs in Cardiomyopathy Research

### 3.1. Disease Modeling

One of the most prominent applications of iPSC technology in the study of cardiomyopathies is its role in disease modeling. This approach involves the generation of patient-specific cardiomyocytes that faithfully recapitulate the cellular phenotypes characteristic of various cardiomyopathy subtypes. By enabling the direct study of disease mechanisms in human-derived cardiac cells, iPSC-based models offer insights that are often inaccessible through traditional animal models or non-cardiac cell lines. A landmark study by Alessandra Moretti et al. in 2010 demonstrated that iPSC-CMs from a patient with long QT syndrome successfully recapitulated hallmark features of the disease, including prolonged action potentials and arrhythmogenic activity [79]. Since then, a wide range of cardiomyopathies and inherited cardiac disorders has been modeled using iPSC-CMs, establishing their value as a versatile platform for studying pathophysiology at the cellular level.

HCM and DCM are the most prevalent inherited cardiomyopathies encountered in clinical practice. Despite the identification of numerous causative mutations, the mechanisms linking genotype to phenotype remain poorly understood [80,81]. This is due to the heterogeneity of mutations, variable penetrance, and limitations of current models. Mouse models often fail to replicate key human features such as septal hypertrophy or left ventricular obstruction seen in HCM [82]. Moreover, human cardiac tissues are scarce and typically represent end-stage disease, limiting mechanistic insights. Therefore, iPSC-CMs have emerged as a transformative model to overcome these challenges. In HCM, iPSC-CMs from patients often show cellular hypertrophy, abnormal calcium handling, and arrhythmic electrical activity in vitro that parallels the patient’s disease. Many studies showed that patient-derived iPSC-CMs with a *MYH7* mutation exhibited increased cell size, upregulation of fetal gene programs, and arrhythmogenic calcium release events, mirroring the HCM phenotype [44,83,84]; correction of the mutation via gene editing normalized these features [85]. Similarly, iPSC models of DCM caused by genetic mutations, such as in *LMNA*, *TNNT2*, or *TTN*, have revealed contractile dysfunction, reduced sarcomere organization, and impaired response to adrenergic stimuli compared to healthy control iPSC-CMs [86,87,88].

iPSC-CM models have significantly advanced our understanding of the pathological mechanisms underlying HCM and DCM. For HCM, abnormal calcium handling consistently emerges as a central pathogenic mechanism, characterized by calcium overload, increased myofilament calcium sensitivity, and arrhythmic events [45,89]. Notably, the study by Timon Seeger et al. demonstrated that patient-derived iPSC-CMs with *MYBPC3* mutations exhibit dysregulation of calcium-related, hypertrophic, stress–response, and structural genes independent of protein haploinsufficiency, challenging prior pathogenic models [90]. Further, gene-specific contractility phenotypes have also been identified, with hypo-contractility observed in the *MYH7*-R453C variant [84] and hypercontractility in the *MYH7*-R403Q and *ACTC1*-E99K variants [91], each of which is responsive to targeted therapeutic interventions. Beyond genetic factors alone, accumulating evidence emphasizes the critical interplay between genetic predisposition and environmental stimuli in exacerbating disease phenotypes. Tanaka et al. demonstrated that iPSC-CMs derived from HCM patients exhibited relatively mild abnormalities under baseline conditions but developed significant cardiomyocyte hypertrophy, pronounced myofibrillar disarray, and increased variability in contractile vectors upon exposure to endothelin-1, highlighting the importance of environmental factors in triggering pathological remodeling [92]. Similarly, in DCM, iPSC-CM models have highlighted impaired sarcomere organization, altered calcium dynamics, defective β-adrenergic signaling, and enhanced apoptotic pathways [93,94]. Molecular investigations have further identified distinct molecular pathways, including disrupted PKA signaling [95], ERK and PDGF pathway dysregulation [96,97], and epigenetic modifications [98,99]. Collectively, these insights underscore the utility of iPSC-CM models, not only for delineating genotype-specific pathological mechanisms but also for identifying novel therapeutic targets, thereby bridging fundamental research with clinical intervention.

Expanding beyond sarcomeric defects, iPSC-CM models have also shed light on the critical role of cytoskeletal proteins in cardiomyopathy pathogenesis. Among these, *DES*, encoding the intermediate filament protein desmin, is essential for preserving cardiomyocyte structural integrity by linking sarcomeres to the nucleus, mitochondria, and plasma membrane. A recent study by Ebrahim et al. (2025) reported a pathogenic DES missense mutation, p.R127P (c.380G>C), segregating in a large family with a high incidence of cardiomyopathy and sudden cardiac death [100]. Functional studies using iPSC-CMs from affected individuals demonstrated that the mutant desmin caused severe filament assembly defects and the formation of cytoplasmic protein aggregates, which persisted even in the presence of wild-type desmin. High-resolution imaging techniques, including confocal and atomic force microscopy, confirmed pronounced ultrastructural disorganization [101,102]. These iPSC-based models of desminopathy underscore the expanding role of cytoskeletal defects in inherited cardiomyopathies and highlights the value of patient-specific platforms in uncovering mechanistic insights and guiding personalized therapeutic strategies.

ACM, which has been historically difficult to study due to its complex fibro-fatty pathology, has been effectively modeled using iPSC-CMs. A pivotal study by Dongrui Ma et al. demonstrated that reprogramming fibroblasts from ARVC patients carrying *PKP2* mutations into iPSCs, followed by cardiomyocyte differentiation, could reproduce hallmark disease features. These included mislocalization and structural abnormalities of desmosomal proteins and a shift toward an adipogenic phenotype [103]. Subsequent studies have further advanced understanding of ACM pathogenesis, revealing key phenotypes such as lipid droplet accumulation, increased expression of adipogenic transcription factors (notably PPAR-γ and PPAR-α), and activation of downstream lipogenic pathways [104,105]. These alterations are often accompanied by cardiomyocyte apoptosis, sodium channel downregulation, and impaired calcium handling. Mechanistic studies have also identified the RhoA/ROCK pathway as critical for maintaining cardiomyocyte identity, with disruptions to the RhoA/MRTF/SRF-signaling circuit leading to nuclear exclusion of transcription factors and lineage switching toward adipocytes [106]. Moreover, mutations in *OBSCN* and *DSG2* genes have been linked to electrophysiological abnormalities and heightened adrenergic sensitivity, implicating ion channel dysfunction in ACM pathophysiology [107,108]. Desmoplakin (*DSP*) missense variants have also been shown to promote pathological degradation and protein insufficiency [109].

Rare cardiomyopathies and multisystem disorders with cardiac involvement have also been effectively modeled using iPSC-CMs. In Danon disease, which is caused by a deficiency of lysosome-associated membrane protein 2 (LAMP2), Kwong-Man Ng et al. demonstrated that iPSC-CMs deficient in LAMP2 recapitulate key pathological features, including lysosomal storage defects and cardiomyocyte hypertrophy, consistent with the clinical cardiac phenotype [110]. Similarly, patient-specific iPSC-CMs from individuals with Fabry and Pompe diseases have reproduced lysosomal accumulation and impaired autophagic flux, offering valuable insights into the molecular pathophysiology of these metabolic cardiomyopathies [111,112]. iPSC lines have also been generated from patients with transthyretin amyloid cardiomyopathy with mutations in the transthyretin gene [113,114]. In addition to genetic forms, iPSC-CMs have been applied to model-acquired cardiomyopathies. Luis Peter Haup et al. exposed healthy iPSC-CMs to doxorubicin, a chemotherapeutic agent, and observed hallmark features of cardiotoxicity, such as oxidative stress, mitochondrial dysfunction, and increased apoptosis, thus establishing a platform for investigating chemotherapy-induced cardiomyopathy [115]. Likewise, ethanol-treated iPSC-CMs have been used to simulate alcoholic cardiomyopathy, revealing impaired calcium handling and contractile dysfunction [116,117].

iPSC-based disease models have been successfully established for a broad spectrum of cardiomyopathies, providing powerful platforms to investigate human cardiac pathophysiology in vitro. These “heart-in-a-dish” systems facilitate the study of disease mechanisms using patient-specific cardiomyocytes and serve as a valuable alternative in contexts where human myocardial tissue is inaccessible or where animal models fail to accurately replicate the complexity of human cardiac diseases. The relevance of these models is further amplified by a recent regulatory shift: Beginning in 2025, the U.S. Food and Drug Administration (FDA) may begin to phase out the requirement for animal testing in the preclinical evaluation of certain drugs and monoclonal antibodies. This landmark change could pave the way for organoid models to play a central role in preclinical research, enabling the discovery and evaluation of therapeutic candidates in a human-relevant context.

### 3.2. Drug Screening and Personalized Medicine

The advent of iPSC-CMs has markedly transformed preclinical drug evaluation by providing a physiologically relevant human cellular model for drug screening, toxicity assessment, and personalized therapeutic approaches. Traditionally, drug development has heavily depended on animal models and non-cardiac cell systems, which frequently fail to accurately predict human-specific cardiac responses due to fundamental differences in cardiac physiology and molecular signaling pathways [118,119]. Human iPSC-CMs bridge this critical gap by enabling direct analysis of drug effects on cardiomyocytes derived from patient-specific genetic backgrounds under highly controlled laboratory conditions [120].

A notable example of integrating iPSC-CMs into the regulatory landscape is the Comprehensive in vitro Proarrhythmia Assay (CiPA) initiative, endorsed by the U.S. Food and Drug Administration (FDA), which incorporates human iPSC-CMs to evaluate the proarrhythmic risks associated with new drug candidates [121]. Specifically, iPSC-CM assays can detect subtle electrophysiological disturbances, including prolongation of action potentials and altered calcium transients, which are indicative of arrhythmogenic potential such as Torsades de Pointes. Thus, CiPA assays utilizing iPSC-CMs enhance the sensitivity of traditional methods such as hERG channel assays and animal models, potentially reducing the occurrence of late-stage drug failures attributed to cardiotoxicity [122].

Recent advances in automated high-throughput screening (HTS) platforms have further expanded the utility of iPSC-CMs in drug discovery, enabling rapid evaluation of extensive chemical libraries against cardiomyocytes with defined genetic mutations. In a seminal study, Yoshinaga et al. performed HTS using iPSC-CMs derived from patients with type 3 long QT syndrome, successfully identifying novel anti-arrhythmic compounds, including analogs of mexiletine, that effectively corrected prolonged action potentials characteristic of the disease [123]. Such precision pharmacology exemplifies a critical shift toward personalized medicine, whereby patient-derived cells inform tailored therapeutic interventions optimized to specific genetic and molecular disease profiles [124]. Similarly, patient-specific iPSC-CMs derived from familial cardiomyopathy cases have been utilized to screen candidate drugs targeting disease-specific phenotypes, including contractile dysfunction or pathological hypertrophy, effectively creating a personalized “clinical trial in a dish” [125,126].

Beyond drug efficacy studies, iPSC-CMs are invaluable for cardiotoxicity testing, addressing a major cause of drug withdrawal from clinical use due to cardiac adverse effects such as arrhythmias, impaired contractility, and structural cardiac damage [127]. Human iPSC-CMs, due to their representation of cardiac ion channels and signaling pathways, effectively reveal drug-induced toxicities that might be missed in non-human models or non-cardiac cells [128]. For instance, Sharma et al. employed high-content imaging of iPSC-CMs to screen kinase inhibitors and chemotherapeutics, revealing distinctive cardiotoxic phenotypes such as disrupted beating patterns, calcium dysregulation, and morphological abnormalities [129]. Further innovation has been achieved by integrating artificial intelligence (AI) approaches into toxicity screening workflows. A recent study by Grafton et al. applied deep learning methods to high-throughput imaging data from iPSC-CMs exposed to diverse chemical libraries, successfully identifying both known and previously unrecognized cardiotoxic compounds, as well as chemical structures predictive of cardiotoxicity [130]. This integration of AI and human-derived cellular platforms represents a significant advancement in predictive toxicology, offering improved detection and understanding of adverse cardiac effects earlier in the drug development pipeline.

Importantly, iPSC-CM platforms allow for exploration of pharmacological responses across diverse genetic backgrounds, which is a critical aspect of personalized medicine. Given the variability in therapeutic responses among patients with identical cardiomyopathies, generating iPSC-CM panels from multiple individuals provides valuable insights into genotype-specific drug responses. Studies have demonstrated that iPSC-CMs harboring different mutations in the same cardiomyopathy-associated gene exhibit distinct responses to standard heart failure medications or experimental therapeutics [131,132]. By screening a panel of patient-specific iPSC-CMs, researchers, and in the future, clinicians, can identify which patient’s cells benefit from a particular drug or which might be harmed. This paradigm has been explored in congenital long QT syndrome: patient-specific iPSC-CMs were used to test various anti-arrhythmic drugs to see which one normalized the electrophysiological parameters best for each individual [133]. Another emerging application is using iPSC-CMs to evaluate gene therapies or biologics on a personalized level—such as assessing the functional rescue efficacy of AAV-mediated gene delivery in vitro before patient treatment [134].

Overall, iPSC-derived cardiomyocytes are increasingly becoming an integral component of modern drug development pipelines. They simultaneously serve as sensitive predictive tools for identifying cardiotoxicity risks and as personalized platforms for discovering and optimizing treatments tailored to individual patient profiles. Looking forward, routine clinical implementation of patient-derived iPSC-CM assays may facilitate more precise and effective management of cardiomyopathies, enabling clinicians to select the safest and most efficacious therapies and significantly enhancing therapeutic outcomes in personalized cardiac care.

### 3.3. Gene Editing and Therapeutic Approaches

Gene editing technologies, particularly CRISPR/Cas9, have revolutionized the field of disease modeling and therapeutic development, especially when integrated with iPSC models of cardiomyopathy. By enabling precise genomic alterations, these tools allow researchers to dissect disease mechanisms and test gene correction strategies in iPSC-CMs. Two primary applications have emerged: (1) the generation of isogenic disease models through the introduction of pathogenic mutations, and (2) the correction of disease-causing mutations for therapeutic validation.

In disease modeling, CRISPR/Cas9 is employed to introduce specific mutations associated with cardiomyopathies into healthy iPSCs. This approach effectively eliminates genetic background variability and enables direct attribution of phenotypic changes to the mutation of interest. A notable example is the study by Hinson et al., in which iPSC-CMs were generated from patients harboring pathogenic mutations in TTN, the gene encoding Titin. To validate the pathogenic role of these variants, the same mutations were precisely introduced into an independent isogenic iPSC line using CRISPR/Cas9. Comparative phenotypic analysis between patient-derived and genome-edited isogenic iPSC-CMs revealed that sarcomere insufficiency due to mutant Titin proteins constitutes a central mechanism in the development of DCM [135]. Additionally, McDermott-Roe et al. investigated the pathogenicity of the *BAG3* (OMIM: 603883) R477H missense variant by introducing it into wild-type iPSC-CMs through CRISPR-based editing. Functional characterization revealed that *BAG3* is critical for maintaining proteostasis in cardiomyocytes. The mutation disrupted this function, thereby elucidating a molecular mechanism for *BAG3*-associated DCM and reinforcing the utility of gene-edited iPSC-CMs for mechanistic cardiovascular research [136].

From a therapeutic standpoint, iPSC-CMs offer a robust and versatile platform for the preclinical evaluation of gene correction strategies, including those with translational potential. CRISPR/Cas9-mediated correction of disease-associated mutations in patient-specific iPSCs not only models the feasibility of autologous cell replacement therapies but also provides critical insights to guide the development of in vivo genome editing approaches. An early demonstration of the utility of CRISPR-Cas9 genome editing in cardiomyopathy modeling involved the correction of a pathogenic *SCN5A* (OMIM: 600163) mutation in patient-derived iPSC-CMs, thereby establishing the causative role of *SCN5A* in arrhythmogenic right ventricular cardiomyopathy (ARVC) [137]. Since then, this approach has been widely adopted for modeling various inherited cardiomyopathies. These include *MYBPC3* mutations in HCM [90], *TBX20* (OMIM: 606061) mutations in left ventricular noncompaction (LVNC) [99], and *RAF1* (OMIM: 164760) mutations implicated in HCM associated with Noonan syndrome [138]. Advancements in genome editing technologies have further enhanced the precision and safety of genetic correction. Next-generation tools such as base editors and prime editors allow for single-nucleotide modifications without inducing double-strand DNA breaks, thereby reducing the risk of unintended genomic damage [139,140,141,142]. A compelling application of base editing was demonstrated in a model of catecholaminergic polymorphic ventricular tachycardia (CPVT), where correction of a splice-site mutation in RYR2 successfully abolished arrhythmogenic calcium transients in iPSC-CMs, underscoring the therapeutic promise of these refined editing platforms [143].

Beyond CRISPR-based genome editing, iPSC-CMs serve as a versatile platform for exploring various gene modulation strategies, such as RNA interference (siRNA/shRNA) and gene overexpression, to investigate therapeutic mechanisms. For example, siRNA-mediated knockdown of MTSS1 in iPSC-CMs derived from patients with DCM due to TTN truncating variants has been shown to enhance contractile function, suggesting a potential therapeutic avenue [144]. iPSC-CMs also provide a human-relevant system to assess the efficacy and tropism of gene delivery vectors, including adeno-associated viruses (AAVs) [145]. In the context of Duchenne muscular dystrophy (DMD)—related cardiomyopathy—patient-derived iPSC-CMs have been utilized to evaluate exon-skipping oligonucleotides and CRISPR-mediated gene editing approaches aimed at restoring dystrophin expression [146,147]. These interventions have demonstrated functional rescue, evidenced by improved contractility and membrane integrity.

Furthermore, gene editing in iPSC-based models offers a valuable tool for identifying novel therapeutic targets. Targeted knockdown or knockout of candidate genes in patient-derived iPSC-CMs enables systematic evaluation of gene function in disease modulation [148]. This functional genomics approach facilitates the discovery of disease modifiers, which may be amenable to pharmacological intervention. For instance, suppression of maladaptive signaling pathways or enhancement of protective mechanisms through genetic perturbation can reveal potential druggable targets [149].

In summary, the integration of iPSC technology with gene editing platforms offers a transformative approach for modeling, understanding, and correcting inherited cardiomyopathies. These methodologies enable precise recreation and rectification of disease phenotypes, supporting their application in preclinical therapeutic development. Data derived from iPSC-CM models have already contributed to the design of early-phase clinical trials for CRISPR-based interventions, including in vivo genome editing for transthyretin amyloidosis and base editing strategies targeting inherited cardiac arrhythmias (Figure 2).

## 4. Challenges and Limitations of iPSC-CM Models in Cardiomyopathy Research and Therapy

While iPSC-derived cardiomyocytes have opened exciting avenues, they come with several challenges that currently limit their utility in certain applications.

### 4.1. Immaturity of iPSC-CMs

A key limitation of iPSC-CMs is their immature phenotype, which more closely resembles fetal or neonatal rather than adult cardiomyocytes. Morphologically, they are smaller, exhibit a rounder shape, lack rod-like geometry, and display disorganized sarcomeres with an absence of transverse tubules [150,151]. Functionally, iPSC-CMs demonstrate spontaneous automaticity, depolarized resting membrane potentials, and impaired calcium handling due to reduced expression of ion channels and calcium-regulatory proteins, leading to diminished contractile strength and slower kinetics [152]. Metabolically, they predominantly rely on glycolysis rather than oxidative phosphorylation, reflecting underdeveloped mitochondrial function relative to mature cardiomyocytes [153]. These features limit their utility in modeling adult-onset cardiomyopathies and in predicting pharmacologic responses that depend on mature electrophysiological or metabolic profiles. Accordingly, advancing the maturation of iPSC-CMs remains a critical focus in the field.

### 4.2. Variability and Modeling Limitations

iPSC lines exhibit notable line-to-line and batch-to-batch variability in their differentiation efficiency and cardiomyocyte phenotypes, stemming from genetic background, epigenetic memory, and reprogramming-related changes [154,155]. Even clones from the same individual can yield differing purity, beating behavior, and ion channel profiles. Protocol inconsistencies across laboratories further compound this variability, challenging reproducibility. While quality control measures and standardized protocols improve consistency, intrinsic biological variability persists. Additionally, prolonged culture may introduce mutations, which may pose safety concerns [156].

Moreover, standard 2D iPSC-CM cultures lack the structural and cellular complexity necessary to model tissue-level cardiac pathologies, such as fibrosis or inflammation [157]. They are predominantly composed of cardiomyocytes and exclude key non-myocyte populations involved in disease progression. To address these limitations, 3D engineered heart tissues (EHTs) and cardiac organoids incorporating supporting cell types have been developed. These models better replicate mechanical stress responses, multicellular interactions, and functional properties like anisotropic conduction, but require further refinement [158]. Nevertheless, they represent essential tools for studying complex cardiac phenotypes beyond the scope of monolayer cultures.

### 4.3. Cost and Technical Expertise

From a practical standpoint, generating and maintaining iPSC lines and differentiating them to cardiomyocytes is labor-intensive and requires specialized expertise [159]. The high cost of culture reagents, specialized equipment (for electrophysiology or high-content imaging), and time (weeks to months for experiments) can be limiting for widespread use.

### 4.4. Ethical Considerations

Compared to embryonic stem cells, iPSCs largely bypass the ethical controversies of using human embryos, since they are derived from adult tissues. However, ethical questions remain in terms of donor consent (for using someone’s cells to create iPSC lines that might persist indefinitely and even be commercialized) and privacy (genetic information in iPSCs) [160]. Patients donating cells for iPSC banks must be informed that their genome will essentially be preserved in a cell line that others might use; issues of ownership and benefit-sharing can arise if, for instance, a therapy derived from someone’s iPSC leads to a profitable product. Ethicists are actively discussing frameworks to handle these concerns. Another ethical aspect is the equitable access to such advanced therapies [161]. Autologous iPSC therapies could be prohibitively expensive, raising concerns that only very wealthy patients or health systems could afford them, which would exacerbate healthcare inequalities. Efforts like allogeneic cell banks aim to create more off-the-shelf solutions that could be cost-reduced over time [162]. Regulatory frameworks for iPSC-derived products remain under active development by agencies such as the U.S. Food and Drug Administration (FDA) and the European Medicines Agency (EMA). These include establishing standards for donor screening, cell characterization, manufacturing practices, and long-term safety monitoring [163].

In general, the clinical translation of iPSC technology for cardiomyopathy is progressing with measured caution. Continued interdisciplinary collaboration among scientists, clinicians, and regulatory bodies is critical. If current challenges are successfully addressed, iPSC-derived cardiomyocytes hold the potential to significantly advance the treatment of severe cardiomyopathies, offering regenerative options beyond current pharmacological or surgical interventions.

## 5. Future Perspectives and Emerging Trends

The integration of iPSC technology into cardiomyopathy research is continually evolving. Several emerging trends and innovative approaches are poised to enhance the utility of iPSC-derived cardiomyocytes and accelerate their translation.

### 5.1. Advances in Maturation Techniques

To overcome the fetal-like phenotype of iPSC-CMs, a range of maturation strategies has been developed. Electrical pacing enhances electrophysiological maturity and upregulates adult ion channels, while mechanical stimulation through cyclic stretching promotes sarcomere organization and contractile strength. Culturing cells on substrates with physiological stiffness further improves morphology and function [164]. Combined electromechanical stimulation in 3D formats has yielded near-adult phenotypes, including T-tubule formation and mature action potentials [165]. Biochemical approaches, including supplementation with thyroid hormone, glucocorticoids, and fatty acid-rich media, drive metabolic maturation and adult gene expression [166]. Co-culture with non-myocytes (e.g., fibroblasts, endothelial cells, or sympathetic neurons) improves structural and functional characteristics, enhancing β-adrenergic responsiveness and calcium handling [167]. Additionally, micro-patterned surfaces and nanomaterials guide cellular alignment, promoting anisotropic contraction and physiological ion channel distribution [168]. While each method offers incremental benefits, integrative approaches combining mechanical, electrical, biochemical, and architectural cues hold the most promise. Developing standardized, scalable protocols for producing mature iPSC-CMs will be pivotal for disease modeling, drug screening, and therapeutic applications.

### 5.2. Integration of Multi-Omics Approaches

Modern single-cell and multi-omics technologies are enhancing the resolution of iPSC-CM characterization. Single-cell RNA sequencing (scRNA-seq) has revealed significant heterogeneity within iPSC-CM cultures, identifying subpopulations at various maturation stages [169,170]. This allows researchers to track developmental trajectories, enrich for mature cells, and benchmark in vitro models against human fetal heart tissue. Beyond transcriptomic profiling, proteomic and metabolomic analyses provide complementary insights into the dynamic changes occurring during cardiomyocyte maturation. These include shifts in metabolic pathways—such as the transition from glycolysis to fatty acid oxidation—and alterations in protein expression and post-translational modifications, including phosphorylation patterns of contractile proteins [171,172]. These analyses have proven instrumental in identifying molecular deficiencies in iPSC-CMs, including the absence or reduced expression of specific ion channel subunits. The integration of transcriptomic, epigenomic, and proteomic datasets has further enabled the identification of critical regulators of the cardiomyocyte phenotype, such as key microRNAs and chromatin modifications that influence gene expression and cellular function [173]. Importantly, these multi-omics approaches have elevated the utility of iPSC-based models in studying cardiomyopathies by allowing systematic interrogation of the molecular consequences of disease-causing mutations. This includes comprehensive profiling of changes in gene expression, protein abundance, and cellular signaling pathways [174,175]. The scalability, reproducibility, and patient-specific nature of iPSC-derived cardiomyocytes make them a powerful platform for high-dimensional omics analyses. Such models not only advance fundamental insights into cardiac biology and disease pathogenesis but also support the development of more precise and individualized therapeutic strategies.

### 5.3. AI in iPSC-Based Research

As iPSC-CM datasets expand in scale and complexity, AI is increasingly employed to extract meaningful insights. High-content imaging techniques, routinely used to capture contraction dynamics, calcium transients, and cellular morphology, generate vast datasets that are particularly amenable to AI-based phenotype classification and pattern recognition [176,177]. Machine learning (ML) algorithms have been successfully applied to integrate these imaging datasets with multi-omics profiles, enabling the development of predictive models of cardiomyocyte maturation, arrhythmogenic risk, and disease susceptibility based on transcriptomic and epigenomic signatures [178]. In parallel, deep learning (DL) approaches have demonstrated superior performance compared to traditional manual assessments, notably in the identification of cardiotoxic compounds through the analysis of subtle motion irregularities captured by video microscopy [130]. These advancements not only improve the sensitivity and specificity of toxicity detection but also facilitate the discovery of novel phenotypic biomarkers associated with cellular dysfunction [179].

Beyond classification and toxicity screening, AI methodologies have been employed to optimize experimental protocols. Reinforcement learning frameworks, for instance, enable iterative optimization of differentiation conditions, including cytokine concentrations and extracellular matrix compositions, thereby enhancing the yield, efficiency, and functional maturity of iPSC-CMs [180]. In the context of personalized medicine, AI holds significant potential for integrating patient-specific clinical, genetic, and iPSC-derived data to predict individual disease susceptibility or therapeutic response [181]. Overall, the integration of AI into iPSC-CM research represents a critical advancement with the potential to transform precision cardiology by enabling more accurate disease modeling, targeted drug discovery, and personalized therapeutic interventions [182].

## 6. Regenerative Medicine and Future Prospect

The ultimate goal of iPSC technology in cardiomyopathy lies in its application to regenerative therapy—restoring myocardial function through the delivery of new cardiomyocytes. In contrast to earlier cell-based approaches, such as bone marrow-derived cell therapies, iPSC-CMs exhibit intrinsic cardiac properties, which enhance their potential for engraftment, long-term survival, and functional integration within damaged myocardial tissue.

Recent innovations have significantly advanced the translational potential of iPSC-based therapies. Biodegradable injectable scaffolds and hydrogels are being developed to deliver iPSC-CMs in a minimally invasive fashion while providing structural and mechanical support to the injured heart [183]. In addition, three-dimensional (3D) bioprinting of personalized cardiac patches incorporating iPSC-derived cardiomyocytes represents a promising approach for customized tissue repair. These technological advances are steadily bridging the gap between preclinical models and clinical application, making iPSC-based cardiac regeneration a tangible therapeutic strategy [184]. Early-phase clinical trials have now begun to assess the safety and feasibility of using iPSC-CMs in patients, particularly those with ischemic heart disease and heart failure [185,186]. Preliminary results from these trials suggest that iPSC-based therapies may contribute to improved myocardial function in patients with advanced heart failure.

However, it is important to emphasize that this therapeutic approach remains in the early stages. The heart is a complex and dynamic organ, and successful regeneration requires overcoming barriers related to cell retention, survival, electromechanical coupling, and the potential risk of inducing arrhythmias [187]. While early case reports and small patient cohorts offer promising signals, robust evidence from large-scale, controlled clinical trials will be essential to validate sustained functional improvements and long-term clinical outcomes, such as enhanced exercise capacity and survival. Despite these challenges, the initiation of clinical trials represents a pivotal step forward in translating iPSC technology into viable regenerative therapies for cardiomyopathy. This progress offers renewed hope that cell-based interventions may eventually complement—or even replace—current treatments such as mechanical circulatory support and heart transplantation for severe patients.

## 7. Conclusions

In conclusion, iPSC-based cardiomyopathy research has elucidated key mechanisms of disease, accelerated the development of tailored treatments, and initiated the first steps towards regenerating human hearts. Continued interdisciplinary collaboration and cautious, rigorous exploration will be essential to overcoming the remaining barriers. The coming years promise not only deeper scientific insights into heart muscle diseases but also tangible clinical advances—potentially transforming cardiomyopathy from a condition managed by symptom control into one that can be fundamentally altered or cured at the cellular level. The role of iPSCs in this transformation is central, marking a new chapter in cardiovascular medicine where patient-specific pluripotent stem cells drive both discovery and therapy. While significant work remains to fully realize this vision, the progress to date is remarkable. In just over a decade since their introduction, iPSCs have moved from an experimental curiosity to the centerpiece of cardiac precision medicine and a beacon of hope for heart regeneration.

## Figures and Tables

**Figure 1 ijms-26-04984-f001:**
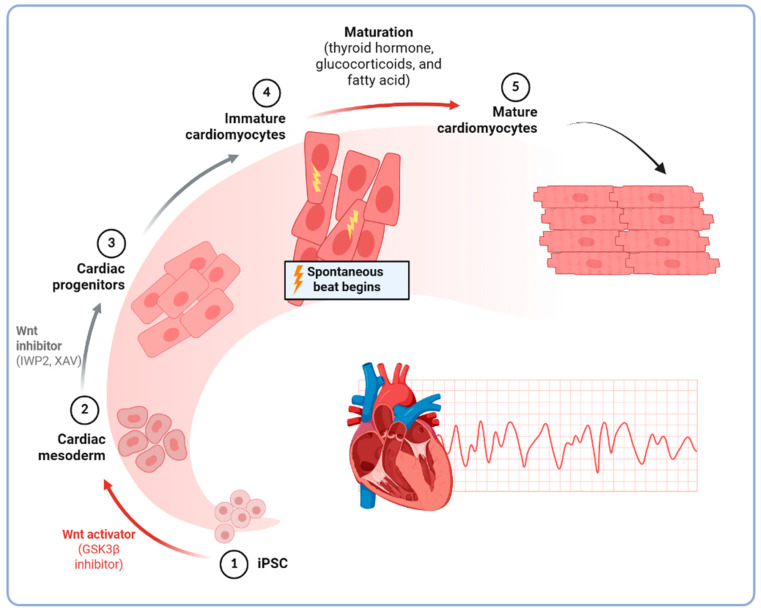
Generation of cardiomyocytes from iPSCs.

**Figure 2 ijms-26-04984-f002:**
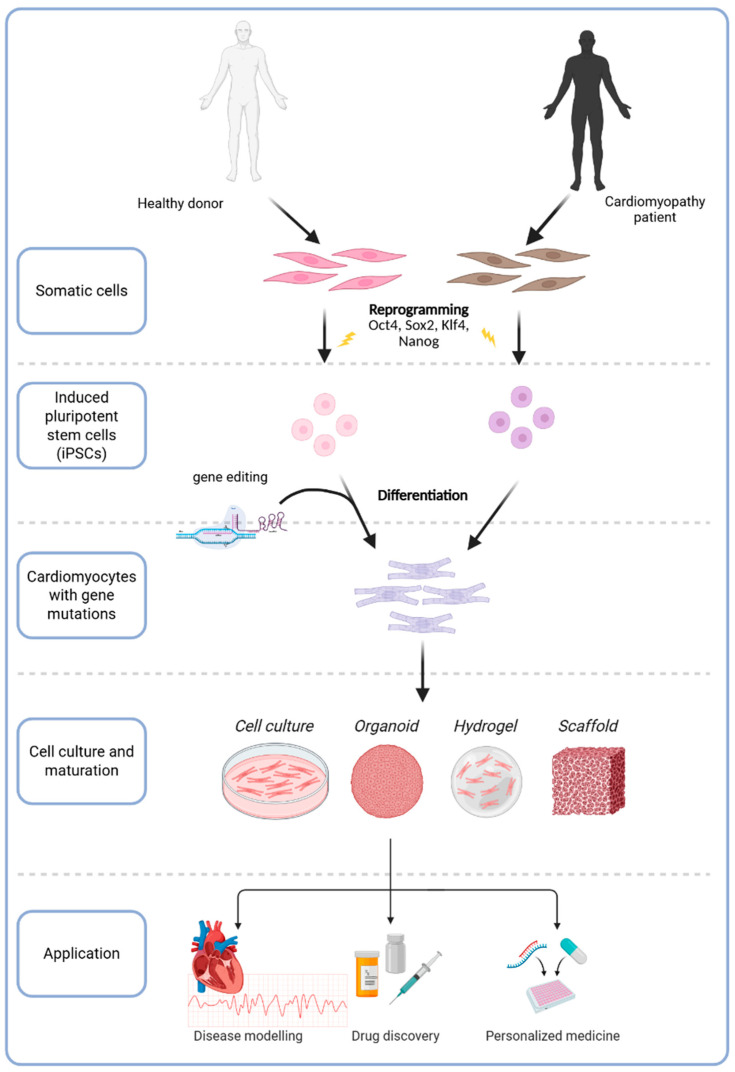
Application of iPSC-based technology in cardiomyopathy.

## Data Availability

No new data were created or analyzed in this study. Data sharing is not applicable to this article.

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
