# Peer review of "Induced Pluripotent Stem Cells in Cardiomyopathy: Advancing Disease Modeling, Therapeutic Development, and Regenerative Therapy"

_ijms, 2025, doi:10.3390/ijms26114984_

Round 1

Reviewer 1 Report

Comments and Suggestions for Authors

In the review article 'Induced Pluripotent Stem Cells in Cardiomyopathy: Advancing Disease Modeling, Therapeutic Development, and Regenerative Therapy' the authors summarize the use of iPSC-CM for research. The topic of this review article is highly relevant and interesting. However, I have several changes, which should be changed before this review article should be published.

  • Within the introduction, I would shortly summarize the genetic landscape of the different cardiomyopathies. The book chapter and the following articles would be really helpful in this context for ARVC and RCM: 'Genetic Insights into Primary Restrictive Cardiomyopathy’ (2022) and ‘Insights Into Genetics and Pathophysiology of Arrhythmogenic Cardiomyopathy’ (Curr Heart Fail Rep. 202). I would shortly explain the major CM genes according to this suggestion and would include adequate references like the suggested articles.
  • Figure 1 could be presented larger. Was this figure generated by Biorender? Than I would include a reference in the figure legend.
  • I would add the DES gene (encoding desmin) in chapter 3. It was shown using iPSC-CM that DES mutations cause an aberrant cytoplasmic aggregation of desmin leading to severe filament assembly defects (see ‘Phenotypic Diversity Caused by the DES Missense Mutation p.R127P (c.380G>C) Contributing to Significant Cardiac Mortality and Morbidity Associated With a Desmin Filament Assembly Defect’ Ebrahim 2025.
  • I would write all human gene names in Italics and would add an OMIM identifier for each gene, when you mention this gene the first time.
  • Could you also increase the size of Figure 2. Was this also generated with Biorender?

In summary, I suggest a major revision changing the five points. However, I am pretty optimistic, that the authors can fix their manuscript at the criticized points.

Author Response

Response to reviewer 1

We would like to express my gratitude for your thoughtful and constructive feedback on our manuscript. Your detailed comments have been invaluable in guiding the revision process. In response to your suggestions, we have meticulously addressed each point to enhance the clarity, completeness, and precision of my work. Below, we outline the revisions made to the manuscript based on your insightful remarks.

Comment: Within the introduction, I would shortly summarize the genetic landscape of the different cardiomyopathies. The book chapter and the following articles would be really helpful in this context for ARVC and RCM: 'Genetic Insights into Primary Restrictive Cardiomyopathy’ (2022) and ‘Insights Into Genetics and Pathophysiology of Arrhythmogenic Cardiomyopathy’ (Curr Heart Fail Rep. 202). I would shortly explain the major CM genes according to this suggestion and would include adequate references like the suggested articles.

Response: We appreciate the reviewer’s suggestion regarding the inclusion of a summary of the genetic landscape of cardiomyopathies. We have revised the Introduction section to include a concise overview of the major genetic contributors to different cardiomyopathy subtypes, with appropriate references to the suggested literature (Ref No.28).

Comment:

  • Figure 1 could be presented larger. Was this figure generated by Biorender? Than I would include a reference in the figure legend.
  • Could you also increase the size of Figure 2. Was this also generated with Biorender?

Response: We confirm that both Figure 1 and Figure 2 were created using BioRender. We have enlarged both figures for improved clarity and added appropriate acknowledgments to BioRender in the Acknowledgment section. Since the figures were self-designed by the authors using BioRender templates, no citation is required in the figure legends.

Comment: I would add the DES gene (encoding desmin) in chapter 3. It was shown using iPSC-CM that DES mutations cause an aberrant cytoplasmic aggregation of desmin leading to severe filament assembly defects (see ‘Phenotypic Diversity Caused by the DES Missense Mutation p.R127P (c.380G>C) Contributing to Significant Cardiac Mortality and Morbidity Associated With a Desmin Filament Assembly Defect’ Ebrahim 2025.

Response: As recommended, we have included a discussion of the DES gene in Chapter 3, highlighting its pathogenic role in cardiomyopathies.

Comment: I would write all human gene names in Italics and would add an OMIM identifier for each gene, when you mention this gene the first time.

Response: We have revised the manuscript to ensure that all human gene names are written in italics and have included the OMIM identifiers when each gene is first mentioned.

Reviewer 2 Report

Comments and Suggestions for Authors

In the manuscript entitled “Induced Pluripotent Stem Cells in Cardiomyopathy: Advancing Disease Modeling, Therapeutic Development, and Regenerative Therapy”, authors highlight the role of induced pluripotent stem cells (iPSCs) in cardiomyopathy research. It discusses how iPSC-derived cardiomyocytes (iPSC-CMs) overcome limitations of traditional models by providing patient-specific human cardiac cells for disease modeling, drug screening and personalized medicine. The article details the generation, differentiation and characterization of iPSC-CMs, describing their applications in understanding disease mechanisms, evaluating drug efficacy and toxicity, and exploring gene editing application. While acknowledging current challenges like iPSC-CM immaturity and variability, the review emphasizes emerging trends such as advanced maturation techniques and multi-omics integration to enhance their potential.

The manuscript could be improved with some issues to take into account.

Maturation – Progesterone and its receptors play an important role in cardiac maturation, influencing metabolic programs and contractile properties. There are several studies on this topic. Very recently however, the role of progesterone/receptor in the reprogramming/differentiation of iPSCs has emerged (10.1007/s12015-024-10776-6). A focus on this last aspect would further enhance the whole manuscript.

Minor English editing and grammatical check (page 8, line 357).

Author Response

Response to reviewer 2

We would like to express my gratitude for your thoughtful and constructive feedback on our manuscript. Your detailed comments have been invaluable in guiding the revision process. In response to your suggestions, we have meticulously addressed each point to enhance the clarity, completeness, and precision of my work. Below, we outline the revisions made to the manuscript based on your insightful remarks.

Comment: Maturation – Progesterone and its receptors play an important role in cardiac maturation, influencing metabolic programs and contractile properties. There are several studies on this topic. Very recently however, the role of progesterone/receptor in the reprogramming/differentiation of iPSCs has emerged (10.1007/s12015-024-10776-6). A focus on this last aspect would further enhance the whole manuscript.

Response: Thank you for this insightful suggestion. We have added a focused discussion on the emerging role of progesterone and its receptor (PR) in the reprogramming and differentiation of iPSCs in Section 2.

Comment: Minor English editing and grammatical check (page 8, line 357).

Response: We have reviewed the manuscript for minor language issues and have corrected the grammatical error noted on page 8, line 357 (page 9, line 400 in the new manuscript), along with other English edits throughout the text.

Round 2

Reviewer 1 Report

Comments and Suggestions for Authors

From my perspective the authors have improved their nice review article at all criticized points. The included figures are interesting and really supportive. Therefore, I suggest to accept this manuscript for publication in IJMS.